# DeepISP: Learning End-to-End Image Processing Pipeline

**Eli Schwartz, Raja Giryes**
School of Electrical Engineering
Tel-Aviv University, Israel
{eliyahus@mail,raja@tauex}.tau.ac.il

**Alex M. Bronstein**
Department of Computer Science
Technion – Israel Institute of Technology, Israel
bron@cs.technion.ac.il

## Abstract

We present DeepISP, a full end-to-end deep neural model of the camera image signal processing (ISP) pipeline. Our model learns a mapping from the raw low-light mosaiced image to the final visually compelling image and encompasses low-level tasks such as demosaicing and denoising as well as higher-level tasks such as color correction and image adjustment. The training and evaluation of the pipeline were performed on a dedicated dataset, the S7-ISP dataset[1], containing pairs of low-light and well-lit images captured by a Samsung S7 smartphone camera in both raw and processed JPEG formats. The proposed solution achieves state-of-the-art performance in objective evaluation of PSNR on the subtask of joint denoising and demosaicing. For the full end-to-end pipeline, it achieves better visual quality compared to the manufacturer ISP, in both a subjective human assessment and when rated by a deep model trained for assessing image quality.

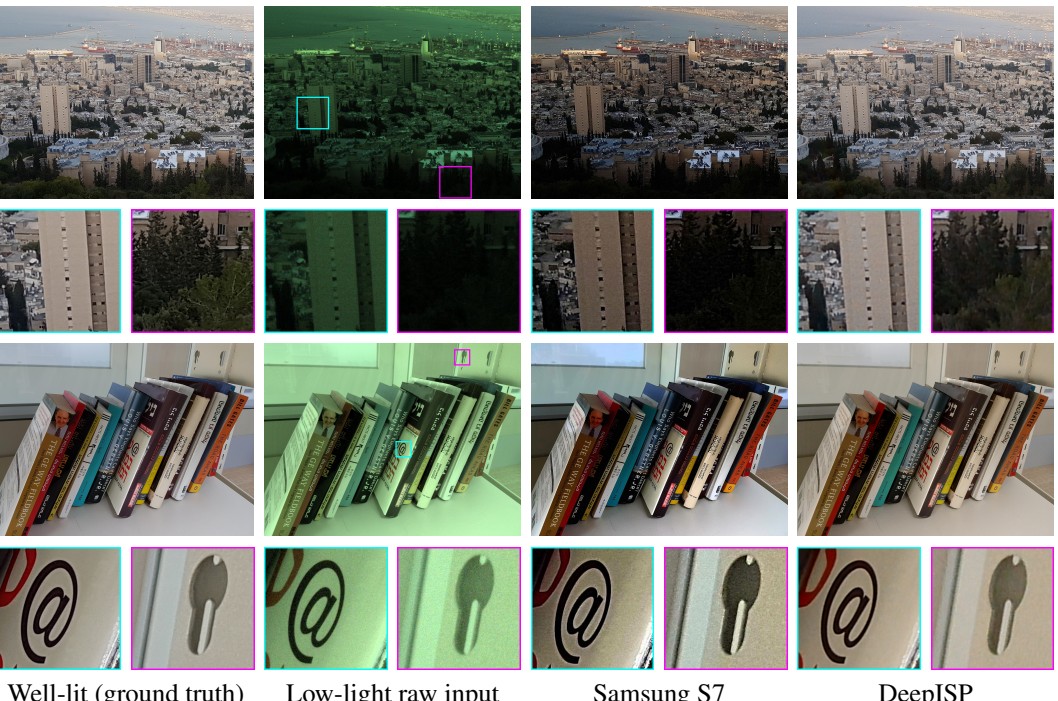

Well-lit (ground truth)    Low-light raw input    Samsung S7    DeepISP

Figure 1: **End-to-end low-light image processing.** From left to right: a ground truth well-lit image, raw input low-light image, output of the Samsung S7 ISP, and of the proposed DeepISP.

---

[1]Dataset and full paper are available on the project page.

# 1 THE DEEPISP NETWORK

We turn to describe now our proposed data-driven solution for the image processing pipeline. The model jointly learns low-level corrections, such as demosaicing and denoising, and higher level global image restoration in an end-to-end fashion. Fig. 2 presents our proposed network architecture for an end-to-end image processing and enhancement, denoted as DeepISP. DeepISP is composed of two stages, depicted in orange and green bordures in the diagram, respectively. The first stage extracts low-level features and performs local modifications. The second one extracts higher level features and performs a global correction. The network is fully convolutional, thus, can accommodate any input image resolution.

**Low-level stage**  The low-level part of DeepISP consists of $N_{ll}$ blocks. Each intermediate block performs convolution with filters of size $3 \times 3$ and stride $1$. Its input and output sizes are $M \times N \times 64$, where $M$ and $N$ are the input image dimensions. The input to the network is a demosaiced RGB image produced by a simple bilinear interpolation in a preprocessing stage.

At each layer, $61$ out of the $64$ channels are standard feed-forward features (left column in the diagram). The other $3$ channels contain a correction for the RGB values of the previous block, i.e., they contain a residual image that is added to the estimation of the previous layer. This design was inspired by DenoiseNet (Remez et al., 2017). Similarly to DenoiseNet, each block produces a residual image. Unlike DenoiseNet, each block also gets as input the current image estimate.

**High-level stage**  The last block at the low-level stage forwards the $61$ feature channels in one path and the currently estimated image ($I$) in another path to the high-level stage of the network. The latter uses the features from the low-level stage for estimating a transformation $W$ that is then applied to the image corrected by the first stage ($I$) to produce a global correction of the image.

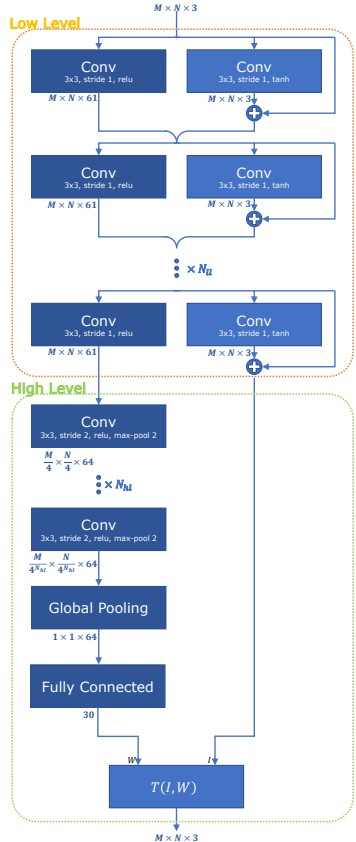

Figure 2: **Proposed network architecture.** The network consists of two stages: Lower level and higher level. The first performs mainly the low-level vision tasks such as denoising and demosaicing. The second stage involves more global processing such as coloring. Yet, both share common features. Layers that output features are colored dark blue. If the output is an image (or residual image) a bright blue is used.

This stage includes a sequence of $N_{hl}$ convolution layers with filters of size $3 \times 3$ and stride $2$. Each layer is followed by a $2 \times 2$ max-pooling. A global mean-pooling is applied to the output of these convolutions, resulting in a single feature vector. This is followed by a fully connected layer that produces the parameters of the transformation $W$. $T(I, W)$ is a function of the pixel's R, G, and B components; it is applied pixel-wise as a linear combination of R, G, and B and the quadratic elements, e.g. $R^2$ and $R \cdot G$.

Our solution for the high-level stage of the network is related to few works Gharbi et al. (2017); Getreuer et al. (2017); Jiang et al. (2017), where a model is also learned to predict a transformation that is then applied to an input image. But they use local transformation and not global like in our work. We found that when combined with the low-level part of the network, using a global model is sufficient for the task at hand and enjoys better convergence and stability.

**Loss**  A commonly used loss for image restoration is the $\ell_2$-distance. While it optimizes mean squared error (MSE), which is directly related to the peak signal-to-noise ratio (PSNR), it leads to inferior results with respect to perceptual quality compared to other loss functions. For learning a full ISP we use a combination of the $\ell_1$ norm and the multi scale structural similarity index (MS-SSIM) to get a higher perceptual quality as suggested by Zhao et al. (2017).

## 2 Joint Denoising and Demosaicing

We start by evaluating our solution on the task of joint denoising and demosaicing. There is a considerable research examining this task and recent studies, e.g., Gharbi et al. (2016); Klatzer et al. (2016), have benchmarked on the MSR demosaicing dataset (Khashabi et al., 2014). For the task of joint denoising and demosaicing, we used the Panasonic images in the MSR dataset for training, and report results for both the Panasonic and Canon test sets (disjoint from the training sets).

|  | Panasonic | | Canon | |
|---|---|---|---|---|
| Method | Linear | sRGB | Linear | sRGB |
| RTF | 37.77 | 31.77 | 40.35 | 33.82 |
| FlexISP | 38.28 | 31.76 | 40.71 | 33.44 |
| DJDD | 38.6 | 32.6 | - | - |
| SEM | 38.93 | 32.93 | 41.09 | 34.15 |
| DeepISP | **39.31** | **33.65** | **41.7** | **35.43** |

Table 1: PSNR for joint denoising and demosaicing. Results are compared to Khashabi et al. (2014); Heide et al. (2014); Gharbi et al. (2016); Klatzer et al. (2016).

As the denoising and demosaicing task requires only local image modifications, we only use the low-level stage of the network. The mosaiced raw image is transformed to an RGB image by bilinear interpolation during the preprocessing stage. Table 1 provides a comparison to other methods. Our proposed technique achieves the best results for joint denoising and demosaicing on both the Panasonic and Canon test sets in the MSR dataset. Compared to the previous state-of-the-art results from Klatzer et al. (2016), our method produces an improvement of $0.38dB$ (linear space) and $0.72dB$ (sRGB space) on the Panasonic test set, and of $0.61dB$ (linear) and $1.28dB$ (sRGB) on the Canon test set.

## 3 Full ISP

**S7-ISP Dataset** For training and assessing the performance of our full pipeline, we generated a dataset of real-world images. For each scene, we captured a JPEG image using the camera fully automatic mode and saved the original raw image too. In addition, we captured a low-light image of the same scene, stored in both JPEG and raw formats. The low-light image was emulated by capturing the same scene with the exact same settings as those chosen by the camera in the automatic mode, except the exposure time that was set to be quarter of the automatic setting.

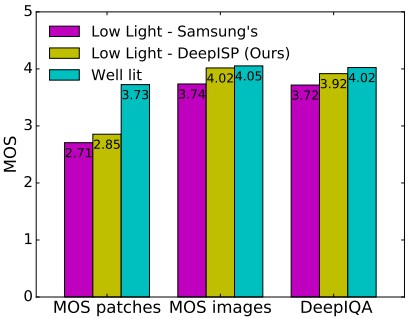

Figure 3: **MOS results for DeepISP.**

**DeepISP Evaluation** The proposed end-to-end model was tested on the challenging task of learning the mapping between low-light raw input images to well-lit JPEG images (produced by the Samsung S7 ISP in automatic setting). To account for the fact that it is difficult to define an objective metric for the full pipeline, we performed a subjective evaluation, generating the mean opinion score (MOS) for each image using Amazon Mechanical Turk to quantitatively assess its quality. A total of 200 ratings have been collected for each image (200 per version of an image, i.e., DeepISP output, Samsung S7 output and the well-lit ground truth): 100 ratings for 10 random patches and additional 100 for the full image. In addition to scoring by humans, we evaluated image quality by a learned model, DeepIQA (Bosse et al., 2016), that was trained to estimate human evaluations. The model output was normalized to the range $[1, 5]$.

Fig. 3 presents the evaluation results. For the patch level, DeepISP MOS is $2.86$ compared to Samsung S7 ISP, which has $2.71$ MOS on the same images. The DeepISP MOS for full images is $4.02$ compared to $3.74$ achieved by Samsung S7 ISP. The former result is only slightly inferior to the MOS $4.05$ that is given to the well-lit images. It is also evident that the visual quality score predicted by DeepIQA (Bosse et al., 2016) corresponds well to the human evaluation with scores of $3.72$, $3.92$ and $4.02$ for the Samsung S7 ISP, DeepISP and the well-lit scene, respectively. Fig. 1 presents a selection of visual results.

**Acknowledgments.** The research was funded by ERC StG RAPID and ERC StG SPADE. The authors thank NVIDIA hardware grant program for donating the Titan X that was used in this research.

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
