# OpenReview forum: "DeepISP: Learning End-to-End Image Processing Pipeline"
_ICLR.cc/2018/Workshop — Reject_

### Official Review · AnonReviewer1 · 2018-03-10
**A Deep Neural Net for End to End Image Signal Processing**

**Rating:** 7
**Confidence:** 4

**Review:**

The paper presents a deep neural net for doing Image Signal Processing (ISP), as done by a digital camera. The network learns the transformations of denoising and demosaicing as a subtask for converting raw low lit mosaiced image into visually pleasing images.  The network has two stages : the first stage sequentially applies convolutional blocks that learn a feature representation and a residual image, the second stage uses the features from the first stage to learn a transform that is finally applied to the image estimate from the first stage. The latter does low local level transformations (denosiing and demosaicing), while the former does a higher level global correction. The first stage network outperforms existing state of the art methods for the problem of joint denoising and demosaicing, evaluated using PSNR. For evaluation of the full model, a new dataset is created by taking images captured by Samsung S7 camera. The full model is successfully able to outdo the industry grade ISP on the Samsung smartphone in a perceptual study (MOS) on AMT and a learnt classifier.

Paper is well written. The work is novel. Neural networks have been applied for the problems of denoising and demosaicing, but this is the first paper that replicates the entire image processing pipeline in a digital camera using a deep neural net. This approach can jointly performs image processing tasks like de-noising, de-mosaicing, color correction, sharpening.

---

### Official Review · AnonReviewer3 · 2018-03-13
**Not deep analysis**

**Rating:** 3
**Confidence:** 5

**Review:**

The work applies a CNN to predict a well lit image given a poorly illuminated image. Authors collect a set of data from poorly-well illuminated image pairs and train CNN to minimize MS-SSIM, and compare their results with the smartphone's built-in processing software. Results are better in MOS terms.

I would not recommend this article for publication. Application and model are well defined and well evaluated (authors even conduct psychophysical experiments). However in my opinion, the most interesting part (Full ISP) does not bring anything new. From a scientific point of view:
- The idea is not new (hundreds of CNN papers end-to-end trained for image processing are available).
- Nothing new is said about why use this particular model/architecture (no comparison with other architectures for complete learning, no analysis of the model trained).
- There is no new information about the training procedure (tips, training time...).
- There are no new ideas about the loss function to use (maybe different perceptual metrics could be used).
- The set of data collected is not well described, is poor (only images with a particular smartphone and fixed settings) and is not really publicly available (you have to register in a system to get it).

From the technical point of view, perhaps a company is interested in the application but:
- The tool is trained and tested with data with only one configuration (smartphone and settings). Can it be used in other devices?
- No information is given on the amount of memory rewired, computational load or time processing.
- No comparison is made with other (maybe simpler) options.

---

> ### Author Response · Authors · 2018-03-23
> **Answer to the reviewer points**
>
> The input of the reviewer is appreciated. Yet, we strongly disagree with the points mentioned in it. In particular, regarding the comparison to previous works and availability of the data. We now turn to address all the points:
>
>
> - The idea is not new (hundreds of CNN papers end-to-end trained for image processing are available).
>
> A: To the best of our knowledge, there is no prior work for learning end-to-end image processing, we would appreciate an example of such paper.
>
> -------------------------------------------------------------------------
>
> - Nothing new is said about why use this particular model/architecture (no comparison with other architectures for complete learning, no analysis of the model trained). - There is no new information about the training procedure (tips, training time...). - There are no new ideas about the loss function to use (maybe different perceptual metrics could be used).
>
> A: We do compare (and outperform) SOTA solutions for the measurable task of joint denoising and demosaicing. While we generally agree more ablation studies regarding architecture, loss function and training procedure can be performed, we believe these don't fit the limited extended abstract format.
>
> -------------------------------------------------------------------------
>
> - The set of data collected is not well described, is poor (only images with a particular smartphone and fixed settings) and is not really publicly available (you have to register in a system to get it).
>
> A: The dataset is available on kaggle.com, which hosts many open datasets. Many researchers have already downloaded and are using it. Here is the link for anyone interested https://www.kaggle.com/knn165897/s7-isp-dataset
> We wonder if the reviewer also thinks that the Imagenet dataset is not really public as it also requires registration.
>
> -------------------------------------------------------------------------
>
> - The tool is trained and tested with data with only one configuration (smartphone and settings). Can it be used in other devices?
>
> A: In the extended version of this short abstract we demonstrate that the tool works also for other lighting conditions. The fact that we treat the phone as a black box, make it easily transferable to any other device.
>
> -------------------------------------------------------------------------
>
> - No information is given on the amount of memory rewired, computational load or time processing.
>
> A: Since it is a fully convolutional model estimating memory and computational costs is trivial.
>
> -------------------------------------------------------------------------
>
> - No comparison is made with other (maybe simpler) options.
>
> A: As mentioned above we do compare to SOTA results for joint denoising and demosaicing. For the full ISP, there is no prior work to compare against and thus we compare against the camera manufacture ISP (which is known to be one of the best on the market).

---

### Decision · Program_Chairs · 2018-03-20
**ICLR 2018 Workshop Acceptance Decision**

**Decision:**

Reject

**Comment:**

Based on the reviews, this paper has not been accepted for presentation at the ICLR workshop. However, the conversation and updates can continue to appear here on OpenReview.